# Factors associated with pregnancy termination in six sub-Saharan African countries

**Rahel Nega Kassa**[1,2]*, **Emily Wanja Kaburu**[1], **Uduak Andrew-Bassey**[1], **Saad Ahmed Abdiwali**[1], **Bonfils Nahayo**[1], **Ndayishimye Samuel**[1], **Joshua Odunayo Akinyemi**[3]

**1** Department of Obstetrics and Gynecology, College of Medicine, Pan African University Life and Earth Sciences Institute, University of Ibadan, Ibadan, Oyo State, Nigeria, **2** Department of Medical-surgical Nursing, School of Nursing, Saint Paul's Millennium Medical College, Addis Ababa, Ethiopia, **3** Department of Epidemiology and Medical Statistics, Faculty of Public Health, College of Medicine, University of Ibadan, Ibadan, Nigeria

* rahelnega208@gmail.com, rahel.nega@sphmmc.edu.et

**Data Availability Statement:** All data sets used/analyzed in this study are uploaded as supplementary information based on request from the journal editor and reviewers. You can also

## Abstract

Pregnancy termination continues to be a leading cause of maternal morbidity and mortality among young women in Africa. The sub-Saharan Africa region has the highest rate of abortion-related deaths in the world, at 185 maternal deaths per 100,000 abortions. The aim of this study is to investigate the factors associated with pregnancy termination among women aged 15 to 29 years in six sub-Saharan African countries. We used secondary data from the most recent Demographic and Health Survey of six sub-Saharan African countries: Kenya, Tanzania, Ethiopia, Burundi, Nigeria, and Rwanda. A total weighted sample of 74,652 women aged 15–29 were analyzed. A multivariable logistic regression model was used to identify the factors associated with pregnancy termination at a p-value < 0.05. Results were presented using adjusted odds ratios (AOR) with 95% confidence interval. The study showed that 6.3% of women aged 15–29 reported pregnancy termination with a higher prevalence rate in Tanzania (8.8%) and lowest in Ethiopia (4%). Highest odds of pregnancy termination occurred among women aged 20–24 as compared to women aged 15–19 in Rwanda (AOR: 4.04, 95%CI 2.05, 7.97) followed by Nigeria (AOR: 2.62, 95% CI 1.99, 3.43), Kenya (AOR: 2.33, 95%CI 1.48, 3.66), Burundi (AOR: 1.99 95%CI 1.48, 2.85), Tanzania (AOR: 1.71 95%CI 1.29, 2.27), and Ethiopia (AOR: 1.69, 95% CI 1.19, 2.42). Women with no education had 4 times higher odds of pregnancy termination compared to women with higher education in Tanzania (AOR: 4.03 95%CI 1.00, 16.13) while women with no education and primary level education were 1.58 times (AOR: 1.58 95% CI 1.17, 2.13) and 1.78 times (AOR: 1.78 95% CI 1.34, 2.37) as likely to terminate pregnancy in Ethiopia. In Tanzania, the likelihood of a pregnancy termination was associated with a relationship to the household head; head (AOR: 3.66, 95% CI (2.32, 5.78), wife (AOR: 3.68, 95% CI 2.60, 5.12), and in-law (AOR:2.62, 1.71, 4.03). This study revealed that a significant number of women had pregnancy termination. Being in the age group of 20–24 & 25–29, having a lower level of education, being a domestic employee and professional, being single/never-in-union, being in the poorest and richer wealth quantile category, and being head, wife,

access all the data sets from the URL https://dhsprogram.com/ by creating an account and preparing a request letter to Measure DHS, the custodian of the online DHS data archive, the Institutional Review Board of ICF International so that they will grant you a permission to access the datasets usen in this study.

**Funding:** The authors received no specific funding for this work.

daughter, and in-law to the household head were the significantly associated with pregnancy termination. Taking these socio-economic factors into consideration by stakeholders and specific sexual education targeted to women aged 15 to 29 would help tackle the problem.

## Introduction

Pregnancy termination occurs when a pregnancy is ended before the 28th week after the last regular menstrual cycle or when the baby is born weighing less than 1000gm [1]. Pregnancy termination is used interchangeably as stillbirth, miscarriage, and forced abortion [2]. Worldwide, about 121 million unwanted pregnancies per year occur and, 61% of these end in abortion. Besides, the World Health Organization(WHO) [3] estimated that one in ten pregnancies results in unsafe abortion, and 68,000 maternal deaths each year because of unsafe abortion [4–6].

The prevalence of pregnancy termination is high in Africa, mostly underreported due to the law, cultural, religious, and societal norms which often pose risks to the health and well-being of women [7]. Hence, pregnancy termination continues to be a leading cause of maternal mortality among young women in Africa. In the sub-Saharan Africa region, 38,000 deaths occur due to unsafe abortions which are preventable [4]. Similarly, there are 5.5 million unsafe abortions performed each year [8]. As of 2019, 6.2 million unsafe abortions occur in the sub-Saharan African region per year. The region has the highest rate of abortion-related deaths in the world, at 185 maternal deaths per 100,000 abortions [9].

As part of sub-Saharan Africa region, one in five maternal deaths in East Africa are due to unsafe abortion [10]. Thirteen percent of maternal fatalities worldwide and up to 25% in some nations are attributable to unsafe abortion [11]. In Africa, unsafe abortions account for more than 40% of all fatalities, making the continent's top cause of maternal mortality [8]. In Nigeria, there is a high prevalence of unplanned pregnancies (81.78%) of which the women often decide to terminate by unsafe means [12]. This is also rampant in Kenya contributing 33.3% of maternal mortality [13, 14].

Unsafe abortion is said to be the second largest cause of maternal fatalities in Tanzania [10]. A significant portion of gynecological admissions involved incomplete abortions, demonstrating that abortion is a significant public health concern in the country. This has led to Tanzania's unacceptable high rate of 454 maternal deaths for every 100,000 live births [15]. The abortion law in Rwanda was amended in 2012 and 2018 on the conditions for women who become pregnant due to rape, child pregnancy, a second degree kinship or if the pregnancy can harm a mother or a fetus [16].

A multilevel analysis of Demographic and Health Survey(DHS) data in selected sub-Saharan African countries revealed that the prevalence of pregnancy termination in Tanzania, Burundi, Rwanda, Nigeria, Kenya and, Ethiopia,19.04%, 17.3%, 17.25%, 15.04%, 11.08% and 9.77% respectively [17, 18]. Pregnancy termination is linked to several issues. Studies carried out in sub-Saharan Africa region discovered that pregnancy termination is related to education level, age, place of living, contraceptive usage and intent, parity, marital status, wealth index, job status, place of residence, and religion. In addition, young women and teenage girls are more likely than older women to terminate their pregnancies [10, 19–21]. The situation was not expected to be statis because there have been programmes targeted at young people are being implemented. Therefore, the aim of this research is to reappraise the level of

pregnancy termination and associated factors among women aged 15 to 29 years in six sub-Saharan African countries.

## Methods

### Study area

Six countries with a DHS conducted not earlier than the year 2014 were selected: five from East Africa and one from West Africa. The selected countries were Nigeria (West Africa), Ethiopia, Tanzania, Kenya, Rwanda, and Burundi (East Africa). These countries were selected because of the high prevalence of abortions in the region [10]. From 1990–1994 to 2015–2019, the share of unintended pregnancies resolved through abortion increased by 44% in both Eastern and Western regions of the continent [19]. Among the selected countries Nigeria and Ethiopia are the two most populous countries in Africa where this menace leads to poor socioeconomic status [20]. Therefore, women of reproductive age in these countries are more likely vulnerable to have a high rate of pregnancy termination. Women aged 15 to 29 were chosen for this study because the age group 15–29 years has peculiar characteristics because they are young, vibrant, most sexually active, and spontaneous in decision-making and are still trying to figure out what the future holds in addition to the high child marriage rate in the study settings. Hence, this age group would most likely to terminate an unplanned pregnancy on occurrence using crude and unsafe means [21–23]. The total number of participants was 74,652. The sample size for each country was: Kenya (17,654), Tanzania (7,512), Ethiopia (9,099), Burundi (10,105), Nigeria (22, 538), and Rwanda (7,744).

### Data source and population

We analyzed cross-sectional data from the Demographic and Health Surveys (DHS) of six countries in sub- Saharan countries. The DHS data were retrieved from the measure online platform. The DHS recode file (Individual response (IR)) women dataset was used for the study. Recent DHS data on pregnancy termination in these selected countries were extracted from respective countries reports with respective year of reports. The data is collected routinely every five years using the same methods and tools in various developing countries. The DHS uses three core questionnaires adapted from the MEASURE DHS project. These questionnaires include the household, women's and men's questionnaires. DHS collects data from household samples using a two-phase stratified cluster design. The sample for all DHS surveys were designed to represent all regions and administrative cities in the countries. The survey participants were selected using stratified and two stage sampling methods: enumeration areas in the first stage and households in the second stage [24–29]. Each region in the selected countries were stratified into urban and rural areas. Then probability proportional to sample size was made. Women aged 15–49 years from selected households were then interviewed. For this study, the DHS dataset was used to extract factors associated with pregnancy termination for women aged 15–29 years. The total number of women aged 15–29 years included in the study from the six countries were 74,652.

### Description of variables

**Dependent/outcome variable.** The outcome variable was "pregnancy termination" derived from the DHS question "Ever had a terminated pregnancy, which was dichotomized as "yes" if the respondent has ever terminated a pregnancy and "no" if the respondent has not terminated pregnancy in the last five years.

**Independent variables.** These include age (15–29), residence (rural, uurban), educational attainment (no education, primary, secondary, and higher education), occupation (professional/technical/managerial, clerical, sales, services, skilled manual, unskilled manual, agricultural and others), wealth index combined (poorest, poorer, middle, richer, and richest), sex of household head (male and female), relationship to household head (head, wife, daughter, daughter-in-law, granddaughter, sister, co-spouse, and other relative) and marital status (Never in union, currently in union/ living with a man, and formerly in union/ living with a man).

## Statistical analysis

Data from Demographic and Health Survey of 6 countries (Burundi, Ethiopia, Kenya, Rwanda, Tanzania and Nigeria) were analyzed using SPSS version 25 statistical software (IBM SPSS Statistics). Frequencies and percentages for each country were computed to describe the demographic characteristics of respondents and outcome variable. Binary and multivariable logistic regression analyses were performed to examine the existence of a relationship between pregnancy termination and the independent variables. Bivariate analysis was conducted for all the independent variables against the outcome variable determining their odds ratio (OR) and p-value. All independent variables that were statistically significant at P-value < 0.25 in the bivariate analysis were entered into the multivariable analysis. The crude odds ratios and adjusted odds ratios (AOR) with their accompanying 95% confidence intervals were used to describe the results.

Variables that had a *p*-value < 0.05 at multivariable analysis were considered as significant factors associated with pregnancy termination among women aged 15–29 years. The goodness of fit of the final model was tested by Hosmer-Lemeshow p-value > 0.05. Multicollinearity between covariates was checked using the variance inflation factor (VIF) and VIF values greater than 2 indicates the existence of multicollinearity.

## Ethics approval

This study was founded on an examination of anonymous secondary data from the DHS in several countries. The survey received ethical approval from the respective countries' National Ethics Committees. Measure DHS, the custodian of the online DHS data archive, Institutional Review Board of ICF International granted permission to use the datasets of the selected countries for this study. The authors had no access to information that could identify individual participants during or after data collection.

## Result

### Socio demographic characteristics of the study participants

In Kenya majority (34.5%) of the participants were between the ages 25–29 while in other five (5) countries: Tanzania (38.7%), Ethiopia (37.2%), Burundi (38.2%), Nigeria (37.5%) and Rwanda (42.1%), majority of the respondents were aged between 15–19 years. In terms of education, Kenya (46.6%), Tanzania (57.4%), Ethiopia (46.4%), Burundi (41.6%) and Rwanda (50.2%) had majority of the respondents with primary education except Nigeria where majority (49.1%) of them had attained secondary education.

Most of the respondents in Kenya (46.8%), Ethiopia (53.1%) and Rwanda (37.8%) were not working, while majority (67.5%) in Burundi and (36.9%) in Tanzania were working in the agriculture sector and 48.0% in Nigeria were working as a clerical staff. Additionally, most of the study participants in Kenya (47.5%), Tanzania (50.1%), Ethiopia (51.7%) and Nigeria

(54.8%) were currently in union or living with a man, while in Burundi and Rwanda 56.0% and 68.9% respectively were singles (never in union).

Respondents were evenly distributed based on the wealth quintile in the countries. In all the countries (Kenya- 56.9%, Tanzania- 62.1%, Ethiopia- 75.8%, Burundi- 85.9%, Nigeria - 55.6% and Rwanda -78.7%) majority of the respondents reside in rural settings. Similarly, majority of the women lived in households headed by males (Kenya -65.8%, Tanzania -77.9%, Ethiopia - 76.7%, Burundi - 73.2%, Nigeria - 83.8% and Rwanda - 68,5%). Lastly, majority in Kenya (36.8%), Tanzania (35.8%), Ethiopia (40.6%) and Nigeria (47.8%) were wives to the household head except for Burundi (44.8%) and Rwanda (50.3%) where the respondents were daughters (Table 1).

## Prevalence of pregnancy termination

Table 2 summarizes the prevalence of pregnancy termination across countries in six sub- Saharan African countries. The overall prevalence of pregnancy termination among these countries was 6.3%. The prevalence of pregnancy termination was highest in Tanzania 8.8% & lowest in Ethiopia which was 4.0%.

## Factors associated with pregnancy termination among women aged 15 to 29 in sub-Saharan Africa

Variables like age groups, highest educational level, employment status, marital status, household wealth quantile, type of residence, sex of household, and relationship to the household head were included in the bivariate regression. All the variables were significantly associated with pregnancy termination among women in Burundi, Ethiopia, Nigeria and Kenya.

In Burundi, the results of bivariate regression showed that the odds of pregnancy termination were 6 times (COR: 5.89, 95% CI 3.93, 8.83) and 11 times (COR: 11.3, 95% CI 7.66, 16.6) higher in age group of 20–24 years and 25–29 compared to age 15–19 years. Women with no education and those with primary school had 2 times (COR: 2.04, 95% CI 1.30, 3.19) and 1.6 times (COR: 1.57, 95% CI 1.13, 2.18) higher chance of pregnancy termination respectively compared to women with higher education. The highest odds of pregnancy termination were found among clerical workers (COR: 1.62 CI 95% 1.23, 2.12), Agricultural /domestic employee (OR: 1.64, 95%CI 1.32, 2.04), and others (OR: 1.48, 95%CI 1.01, 2.17) compared to those not working.

Besides, women who were never-in-union were less likely to terminate a pregnancy compared to formerly in union/living with a man (COR: 0.02, 95%CI 0.01, 0.03) as well women currently in union/living with a man were 1.6 times (COR: 1.63, 95%CI 1.18, 2.26) more likely to terminate a pregnancy than those women formerly in union/living with a man. The odds of pregnancy termination were 2 times (COR: 2.1, 95% CI 1.59, 2.87) higher among women living in rural areas compared to women living in urban areas. Women, whose household head were male, were 1.9 times (COR: 1.93, 95% CI 1.57, 2.39) more likely to terminate a pregnancy compared to those living in the households headed by female.

In Ethiopia, young women in age group 20–24 and 25–29 had higher chances of pregnancy termination respectively compared to women aged between 15–19. The odds of pregnancy termination among women who didn't have education (COR: 2.2 ,95%CI 1.35, 3.62) and who attended only primary school (COR: 1.33, 95%CI .82, 2.19) and higher compared to women who attended higher education. Similarly, professional women (COR: 1.68, 95%CI 1.28, 2.19) and women who engaged in agricultural activities (COR: 1.47, 95%CI 1.12, 1.95) had higher odds of pregnancy termination respectively than women who didn't have work. Women who

**Table 1. Sociodemographic characteristics of women within the age group of 15 to 29 in six sub-Sahara African countries.**

| Variable | Kenya | Tanzania | Ethiopia | Burundi | Nigeria | Rwanda |
|---|---|---|---|---|---|---|
| | (n = 17,654) | (n = 7,512) | (n = 9,099) | (n = 10,105) | (n = 22,538) | (n = 7,744) |
| Age in five years group | | | | | | |
| 15–19 | 5820 (33.0) | 2904(38.7) | 3381(37.2) | 3859(38.2) | 8448 (37.5) | 3258 (42.1) |
| 20–24 | 5735 (32.5) | 2483(33) | 2762(30.4) | 3244(32.1) | 8448 (30.3) | 2414(31.2) |
| 25–29 | 6100 (34.5) | 2125 (28.3) | 2957(32.5) | 3002(29.7) | 7255 (32.2) | 2073(26.8) |
| Highest educational level | | | | | | |
| No education | 905 (5.1) | 765(10.2) | 2723(29.9) | 2351(23.3) | 6946 (30.8) | 221(2.7) |
| Primary | 8221 (46.6) | 4313(57.4) | 4220(46.4) | 4208(41.6) | 2472 (11.0) | 3888(50.2) |
| Secondary | 6536 (37.0) | 2335(31.1) | 1510(16.6) | 3421(33.8) | 11055 (49.1) | 3344(43.2) |
| Higher | 1993 (11.3) | 99(1.3) | 646(7.1) | 126(1.2) | 2065 (9.2) | 301(3.9) |
| Employment status | | | | | | |
| Not working | 3879 (46.8) | 2452(32.6) | 4828(53.1) | 2138(21.2) | N/A | 2929(37.8) |
| Professional/technical | 634 (7.6) | 205(2.7) | 1522(16.7) | 62(0.6) | 1050 (8.4) | 123(1.6) |
| Clerical | 51 (0.3) | 49(0.6) | 94(1.0) | 22(0.2) | 6007 (48.0) | 70(0.9) |
| Agricultural/domestic employee | 2349 (28.4) | 2770(36.9) | 1597(17.6) | 6821(67.5) | 2828 (22.0) | 1899(24.5) |
| Services | 848(10.2) | 740(9.9) | 391(4.3) | 540(5.3) | 1627 (13.0) | 1154(14.9) |
| Skilled manual | 33(0.2) | 269(3.6) | 300(3.3) | 58(0.6) | 932 (7.4) | 178 (2.3) |
| Unskilled manual | 491(2.8) | 1027(13.7) | 124(1.4) | 41(0.4) | 16 (0.1) | 1391(18.0) |
| Others | N/A | N/A | 244(2.7) | 423(4.2) | 52(0.4) | N/A |
| Marital status | | | | | | |
| Never in union | 8132 (46.1 | 3157 (42.0) | 3844(42.2) | 5659(56.0) | 9726(43.2) | 5335(68.9) |
| Currently in union/living with a man | 8383 (47.5 | 3763 (50.1) | 4700(51.7) | 4009(39.7) | 12349(54.8) | 2121(27.4) |
| Formerly in union/living with a man | 1139 (6.5 | 592 (7.9) | 555(51.7) | 438(4.3) | 463 (2.1) | 288(3.7) |
| Household wealth quantile | | | | | | |
| Poorest | 2701 (15.3) | 1236 (16.5) | 1432(15.7) | 1769(17.8) | 3867(17.2) | 1349(17.4) |
| Poorer | 3121(17.7) | 1231 (16.4) | 1644(18.1) | 1975(19.5) | 4545 (20.2) | 1443(18.6) |
| Middle | 3364 (19.1) | 178 (17.0) | 1659(18.2) | 2070(20.5) | 4568 (20.3) | 1411(18.2) |
| Richer | 3738 (21.2) | 1623 (21.6) | 1739(19.1) | 1990(19.7) | 918 (21.8) | 1566(20.2) |
| Richest | 4730 (26.8) | 2144 (28.5) | 2625(28.8) | 23(22.8) | 4639 (20.6) | 1975(25.5) |
| Type of residence | | | | | | |
| Urban | 7601 (43.1) | 2843 (37.9) | 2200(24.2) | 1424(14.1) | 10000 (44.4) | 1651(21.3) |
| Rural | 10054(56.9) | 4668 (62.1) | 6900(75.8) | 8682(85.9) | 12538 (55.6) | 6093(78.7) |
| Sex of household | | | | | | |
| Male | 11621(65.8) | 5852 (77.9) | 6981(76.7) | 7395(73.2) | 18876(83.8) | 5304(68.5) |
| Female | 6033 (34.2) | 1660 (22.1) | 2119(23.3) | 2710(26.8) | 3662(16.2) | 2440(31.5) |
| Relationship to the household head | | | | | | |
| Head | 2299 (13.0) | 337(4.5) | 763(8.4) | 753(7.5) | 1051 (4.7) | 461 (6.0) |
| Wife | 6500 (36.8) | 2686(35.8) | 3693(40.6) | 3362(33.3) | 10773(47.8) | 1810(23.4) |
| Daughter | 5401 (30.6) | 2150(28.6) | 31301(36.3) | 4531(44.8) | 7747 (34.4) | 3892(50.3) |
| In-law | 493 (2.8) | 526(7.0) | 272(3.0) | 50(0.5) | 459 (2.0) | 106(1.4) |
| Other relative | 2125(12.0) | 978(13.0) | 761(8.4) | 1010(10.0) | 2158 (9.6) | 846(10.9) |
| Not related | 836(4.7) | 835(11.1) | 309(3.4) | 398(3.9) | 350 (1.6) | 629(8.1) |

N/A: not estimated due to small sample

were living in the rural area had 1.3 times (COR: 1.29, 95% CI 0.99, 1.67) higher odds of pregnancy than their urban counterparts.

**Table 2. Prevalence of pregnancy termination among women aged 15–29 in six sub-Sahara African countries.**

| Country | Survey year | Study participants (n = 74, 652) | Weighted % | Pregnancy termination % | |
|---|---|---|---|---|---|
| | | | | No (%) | Yes (%) |
| Kenya | 2014 | 17,654 | 23.65 | 94.0 | 6.0 |
| Tanzania | 2016 | 7,512 | 10.06 | 91.2 | 8.8 |
| Ethiopia | 2016 | 9,099 | 12.19 | 96.0 | 4.0 |
| Burundi | 2016/17 | 10,105 | 13.54 | 93. | 6.6 |
| Nigeria | 2018 | 22,538 | 30.19 | 92.4 | 7.6 |
| Rwanda | 2019/20 | 7,744 | 10.37 | 95.5 | 4.5 |
| Total | | 74,652 | 100 | 93.75 | 6.25 |

Except for the type of residence and sex of the household all variables were found to be associated with pregnancy termination among women in Tanzania in the bivariate analysis. Similarly, with the exception of the type of residence, all variables were found to be associated with pregnancy termination among women in Rwanda (Table 3).

Multicollinearity between covariates was assessed using the variance inflation factor (VIF). Almost in all countries, VIF values were less than 2, indicating that there was no multicollinearity between the independent variables (Table 4).

After running multivariate logistic regression analysis, except for type of residence and sex of household, the rest variables were significantly associated with the outcome variable in at least one of the six countries. In the adjusted model, age groups, highest educational level, employment status, marital status, household wealth quantile, and relationship to the household were found to be predictors of pregnancy termination (Table 5).

Highest odds of pregnancy termination occurred among women aged 20–24 as compared to women aged 15–19, in Rwanda (AOR: 4.04, 95%CI 2.05, 7.97) followed by Nigeria (AOR: 2.62 95%CI 1.99, 3.43), Kenya (AOR: 2.33, 95%CI 1.48, 3.66), Burundi (AOR: 1.99, 95%CI 1.24, 3.19), Tanzania (AOR: 1.71 95%CI 1.29, 2.27), and Ethiopia (AOR: 1.69, 95% CI 1.19, 2.42). Similarly, women who were in the age group of 25–29 were more likely to have a higher chance of pregnancy termination compared to women aged 15–19 in Rwanda (AOR: 5.64, 95%CI 2.82, 11.25) followed by Kenya (AOR: 3.5395%CI2.24, 5.57), Nigeria(AOR: 3.2 95%CI 2.44, 4.20), Tanzania (AOR: 2.75 95%CI 2.07, 3.65), Burundi (AOR: 2.54 95%CI 1.58, 4.06), and Ethiopia (AOR: 2.34, 95% CI 1.64, 3.31)

Women with no education had 4 times higher chance of pregnancy termination compared to women with higher education in Tanzania (AOR: 4.03 95%CI 1.00, 16.13) while in Ethiopia, women with no education and primary education were 1.58 times (AOR: 1.58 95% CI 1.17, 2.13) and 1.78 times (AOR: 1.78 95% CI 1.34, 2.37) more likely to terminate pregnancy respectively. Pregnancy termination was associated with employment status in Tanzania, Ethiopia, Nigeria, Kenya, and Burundi which was 2 times (AOR: 2.19, 95% CI 1.59, 2.99) higher in women working in services in Tanzania compared to those who were unemployed. Women working in agricultural activities in Burundi were (AOR: 1.58, 95%CI 1.07, 2.34) more likely to terminate pregnancy compared to those who were not working. In Ethiopia, professional (AOR: 1.76, 95% CI 1.46, 2.11) and, women engaged in services (AOR: 1.68, 95% CI 1.18, 2.42) were more likely to terminate pregnancy respectively compared to unemployed. Women who were engaged in agricultural activities were 41% (AOR: 0.59 95% CI 0.46, 0.76) less likely to perform pregnancy termination in Nigeria compared to unemployed women.

Marital status was significantly associated with pregnancy termination in all countries included in the study. Women who were never-in-union were less likely to terminate

**Table 3. Bivariate analysis to show association of sociodemographic variables with pregnancy termination among women aged 15 to 29 in six sub-Sahara African countries.**

| Variable | Kenya | Tanzania | Ethiopia | Burundi | Nigeria | Rwanda |
|---|---|---|---|---|---|---|
| | (COR (95%CI)) | (COR (95%CI)) | (COR (95%CI)) | (COR (95%CI)) | (COR (95%CI)) | (COR (95%CI)) |
| Age in five years group | | | | | | |
| 15–19 | 1 | 1 | 1 | 1 | 1 | 1 |
| 20–24 | 4.45(1.38, 14.33) * | 3.52(2.72,4.56) * | 4.19(2.92,6.01) * | 11.60(7.52,17.90) * | 4.34 (3.63,5.14) * | 11.7(6.56, 20.86 * |
| 25–29 | 3.23(1.02,10.28) * | 7.06(5.52, 9.02) * | 7.50(5.37,10.7) * | 25.57(16.74,36.07) * | 6.75 (5.71,7.97) * | 30.21(17.21,53.02) * |
| Highest educational level | | | | | | |
| No education | 1.60(1.06,2.42) * | 4.45(1.38,14.33) * | 2.0(1.38,3.14) * | 19.07(2.65,136.9) * | 1.37 (1.13,1.67) * | 2.65(1.19,5.93) * |
| Primary | 1.20(0.84,1.72) | 3.24(1.02,10.28) * | 1.29(0.85,1.95* | 10.76(1.50,77.21) * | 1.56 (1.23,1.94) * | 1.46(0.76,2.79) * |
| Secondary | 0.75(051,1.11) | 1.85(0.58,5.931) | 0.61(0.37,1.03) | 2.27(0.31,16.53) | 0.84 (0.69,1.01) | 1.14(0.59, 2.20) |
| Higher | 1 | 1 | 1 | 1 | 1 | 1 |
| Employment status | | | | | | |
| Not working | 1 | 1 | 1 | 1 | 1 | 1 |
| Professional/technical | 0.59(0.39, 0.90) * | 1.77(1.04,3.00) * | 1.5(1.17,1.93) * | 6.02(2.59,14.01) * | 0.66 (0.34,1.29) | 2.32(0.98,5.47) * |
| Clerical | 0.95(0.57,1.59) | 0.82(.197,3.42) | 1.07(0.43,2.66) | 6.28(1.83, 21.54) * | 0.78 (0.61,0.98) | 2.79(0.98,7.91) * |
| Agricultural/domestic employee | N/A | 2.28(1.84,2.81) * | 1.26(0.94,1.69) * | 5.19(3.67,7.33) * | 1.02 (0.8,1.23) | 3.17(2.31,4.37) * |
| Services | 1.17(0.75,1.82) | 2.33(1.71,3.17) * | 1.1(0.64,1.39) | 4.62(2.91,7.327) * | 1.03 (0.76, 1.40) | 2.45(1.70,3.54) * |
| Skilled manual | 1.22(0.78,1.92) | 1.86(1.23,2.80) * | 0.99(0.53,1.83) | 7.69(3.09,7.33) * | 0.70 (0.25, 1.95) | 2.25(1.06, 4.78) * |
| Unskilled manual | 1.17(0.72,1.90) | 2.52(1.94,3.26) * | 0.67(0.82,0.33) | 3.69(0.85,15.98) * | NA | 3.05(2.17, 4.28) * |
| Others | 0.69(0.09,5.31) | N/A | N/A | 0.87(0.36,2.08) | NA | NA |
| Marital status | | | | | | |
| Never in union | 0.08(0.05, 0.14) * | 0.10(0.07,0.15) * | 0.48(0.02,0.09) * | 0.017(0.09,0.033) * | 0.13 (0.09,0.18) * | 0.07(0.04,0.12) * |
| Currently in union/living with a man | 1.08 (0.78,1.50) | 0.93(0.72,1.19) | 1.71(0.82,1.67) | 1.43 (1.035,1.99) * | 1.20 (0.88,1.62) | 1.63(1.05,2.53) * |
| Formerly in union/living with a man | 1 | 1 | 1 | 1 | 1 | 1 |
| Household wealth quintile | | | | | | |
| Poorest | 0.97(0.72, 1.30) | 1.25(0.96, 1.62) | 1.49(1.16, 1.9) * | 1.91 (1.46, 2.49) * | 1.35 (1.14, 1.61) * | 1.59(1.14, 2.22) * |
| Poorer | 1.04(0.77, 1.40) | 1.38(1.07, 1.78) * | 1.1(0.79, 1.54) | 1.89 (1.45, 2.45) * | 1.25 (1.05, 1.48) * | 1.32(0.94, 1.84) * |
| Middle | 0.93(0.68, 1.27) | 1.48(1.16, 1.89) * | 1.26(0.91,1.74) * | 1.97 (1.53, 2.55) * | 1.23 (1.04, 1.45) | 1.35(0.97, 1.90) * |
| Richer | 1.01(0.74, 1.37) | 1.29(1.02, 1.62) | 0.69(0.47, 1.03) | 1.52 (1.16, 1.98) * | 1.13 (0.95, 1.34) | 1.15(0.81, 1.64) |
| Richest | 1 | 1 | 1 | 1 | 1 | 1 |
| Type of residence | | | | | | |
| Urban | 1.34(0.94, 1.37) * | 0.87(073,1.03) | 1.18(0.96,1.47) * | 0.60(0.48, 0.75) * | 0.88 (0.80, 0.98) * | 1.04(0.81, 1.33) |
| Rural | 1 | 1 | 1 | 1 | 1 | 1 |
| Sex of household head | | | | | | |
| Male | 1.32(1.08, 1.61) * | 1.1(0.90, 1.34) | 1.78(0.94,1.48) * | 1.84 (0.49, 2.28) * | 1.59 (1.36, 1.86) * | 2.11(1.59, 2.78) * |
| Female | 1 | 1 | | 1 | 1 | 1 |
| Relationship to the household | | | | | | |
| Head | 5.04(1.84, 13.82) * | 3.66(2.31, 5.78) * | 3.89(1.99,7.60) * | 9.71 (4.20, 22.45) * | 1.95 (1.08, 3.53) | 7.63(3.52, 16.52) * |
| Wife | 5.96(2.20, 16.15) | 3.68(2.60, 5.21) * | 4.25(2.24,8.05) * | 14.89 (6.62, 39.49) * | 2.30 (1.31, 4.04) * | 13.57(6.67, 27.60) * |
| Daughter | 0.88(0.31, 2.49) | 0.79(0.53, 1.18) | 0.57(0.28,1.16) * | 0.60 (0.25, 1.45) | 0.31 (0.17, 0.55) * | 0.85(0.39, 1.82) |
| In-law | 5.57(1.89, 16.43) | 2.63(1.71, 4.03) * | 3.64(1.57,8.46) * | 1.73 (0.20, 14.67) | 1.67 (0.88, 3.17) | 0.80(0.10, 6.53) |
| Other relatives | 0.98(0.107, 8.88) | 1.51(0.99,2.29) | 0.53(0.22,1.29) * | 1.45 (0.58, 3.66) | 0.58 (0.31, 1.06) | 0.69(0.25, 1.92) |
| Not related | 1 | 1 | 1 | 1 | 1 | 1 |

COR crude odds ratio, CI confidence interval

*Shows statistically significant association at p-value < = 0.25

**Table 4. Table that shows multicollinearity between covariates.**

| Variables | Kenya | | Tanzania | | Ethiopia | | Burundi | | Nigeria | | Rwanda | |
|---|---|---|---|---|---|---|---|---|---|---|---|---|
| | T | VIF | T | VIF | T | VIF | T | VIF | T | VIF | T | VIF |
| Age in five years group | .59 | 1.67 | .68 | 1.45 | .72 | 1.38 | .62 | 1.59 | .71 | 1.39 | .62 | 1.59 |
| Highest education level | .72 | 1.38 | .72 | 1.37 | .64 | 1.54 | .66 | 1.50 | .58 | 1.72 | .76 | 1.31 |
| Employment status | .85 | 1.17 | .89 | 1.11 | .99 | 1.01 | .93 | 1.07 | .98 | 1.01 | .86 | 1.15 |
| Marital status | .60 | 1.65 | .65 | 1.52 | .68 | 1.45 | .49 | 2.02 | .60 | 1.66 | .58 | 1.72 |
| Household wealth quantile | .50 | 1.98 | .74 | 1.34 | .55 | 1.81 | .65 | 1.52 | .51 | 1.96 | .71 | 1.40 |
| Type of residence | .61 | 1.63 | - | - | .54 | 1.84 | .70 | 1.41 | .67 | 1.48 | - | - |
| Sex of household | .97 | 1.02 | - | - | .93 | 1.06 | .97 | 1.02 | .92 | 1.08 | .97 | 1.02 |
| Relationship to the household | .81 | 1.2 | .85 | 1.17 | .83 | 1.19 | .62 | 1.61 | .83 | 1.20 | .69 | 1.44 |

T: tolerance; VIF: variance inflation factor

pregnancy compared with formerly married women in Burundi (AOR: 0.024, 95%CI 0.01, 0.06), Ethiopia (AOR: 0.11, 95% CI 0.07, 0.17), Rwanda (AOR: 0.17, 95% CI 0.09, 0.30), Kenya (AOR: 0.17 95% CI 0.10, 0.28), and Tanzania (AOR: 0.26, 95%CI 0.18, 0.38). However, in Nigeria, women who were currently in union or living with a man were 5.35 times more likely (AOR: 5.35 95% 4.1, 6.98) to terminate pregnancy compared to women who were formerly in union.

As shown in Table 5, household wealth quintile was significantly associated with pregnancy termination only among women in Kenya, Burundi and Ethiopia. Specifically, pregnancy termination was less likely to be performed among women in the poorest wealth quantile compared to women in the richest wealth quantile in Kenya and Burundi, (AOR: 0.58, 95%CI 0.38, 0.89), and (AOR: 0.62, 95%CI 0.44, 0.88) respectively. The study also found that women who were in richer household wealth quantile had 32% (AOR: 0.68, 95%CI 0.49, 0.96) lesser odds of having a pregnancy termination compared to those who were in the richest wealth quintile category.

Relationship to the household head was also significantly associated with pregnancy termination among women in Burundi and Nigeria. In Burundi, women who were household head (AOR: 0.17, 95% CI 0.05, 0.53), daughter (AOR: 0.33, 95% CI 0.11, 96) and in-law (AOR: 0.052, 95% CI 0.006, 0.48) were less likely to terminate pregnancy compared to women who were not related to the household head. Similarly, women in Nigeria who were daughter to the household head (AOR: 0.46 95%CI 0.25, 0.85) had lesser odds of pregnancy termination than those women who were not relatives.

## Discussion

This study aimed to determine the prevalence of pregnancy termination and its associated factors among women in aged of 15–29 years in selected sub-Sahara African countries. The overall prevalence of pregnancy termination in the current study was 6.3% with higher prevalence in Tanzania (8.8%) and lowest in Ethiopian (4.0%). This finding is consistent with prior evidence from Nigeria, and Ethiopia [7, 30–32]. However, it is higher compared to the study done in European countries, United States of America and Norway respectively [33–35]. The low socioeconomic status among women in the sub-Sahara Africa countries might lead to have unplanned pregnancy and ultimately pregnancy termination. Barriers to accessing and using effective sexual and reproductive healthcare might also exist that might lead to high rate of unplanned pregnancies as well as abortions in the non-developed countries compared to the well-developed ones [36–39].

**Table 5. Factors associated with pregnancy termination among women aged 15 to 29 in six sub-Sahara African countries.**

| Variable | Kenya | Tanzania | Ethiopia | Burundi | Nigeria | Rwanda |
|---|---|---|---|---|---|---|
| | (AOR (95%CI)) | (AOR (95%CI)) | (AOR (95%CI)) | (AOR (95%CI)) | (AOR (95%CI)) | (AOR (95%CI)) |
| Age in five years group | | | | | | |
| 15–19 | 1 | 1 | 1 | 1 | 1 | 1 |
| 20–24 | 2.33(1.48, 3.66) * | 1.71 (1.29, 2.27) * | 1.69(1.19, 2.42) * | 1.99(1.24, 3.19) * | 2.62(1.99, 3.43) | 4.04(2.05, 7.97) * |
| 25–29 | 3.53(2.24, 5.57) * | 2.75 (2.07, 3.65) * | 2.33(1.64, 3.31) * | 2.54(1.58, 4.06) * | 3.2 (2.44, 4.20) * | 5.64(2.8, 11.25) * |
| Highest educational level | | | | | | |
| No education | 1.44 (0.83, 2.49) | 4.03 (1.00, 16.13) * | 1.58(1.17, 2.13) * | 5.18(0.66, 40.35) | 1.03 (0.77, 1.39) | 1.19(0.50, 2.81) |
| Primary | 1.54 (1.04, 2.29) * | 4.13 (1.05, 16.24) * | 1.78(1.34, 2.37) * | 5.14(0.66, 39.89) | 1.34 (0.99, 1.79) | 1.35(0.68, 2.69) |
| Secondary | 1.29(0.62, 1.45) | 3.08 (0.90, 10.49) | 0.99(.56, 1.68) | 2.94(0.382, 22.61) | 1.140 (0.89, 1.46) | 1.48(0.76, 2.88) * |
| Higher | 1 | 1 | 1 | 1 | 1 | 1 |
| Employment status | | | | | | |
| Not working | 1 | 1 | 1 | 1 | 1 | 1 |
| Professional/technical | 0.91 (0.5, 1.39) | 1.08 (0.60, 1.96) | 1.76(1.47, 2.12) * | 1.95(0.7, 5.14.) | 0.58 (0.29, 1.12) | 0.74(0.27, 2.04) |
| Clerical | N/A | 0.91 (0.21, 3.97) | 1.92(.71, 5.18) | 1.59(0.84, 13.762) | 0.68 (0.54, 0.86) * | 1.55(0.541, 4.44) |
| Agricultural/domestic employee | 0.67 (0.49,0.90) * | 1.33 (1.02, 1.73) * | .99(.73,1.4) | 1.58(1.07, 2.34) * | 0.86(0.66,1.12) | 1.18(0.83, 1.68) |
| Services | 1.13 (0.87, 1.46) | 2.19 (1.59, 2.99) * | 1.69(1.18, 2.42) * | 1.31(0.79, 2.17) | 0.59 (0.46, 0.76) * | 1.45(0.988, 2.14) |
| Skilled manual | 0.55 (0.072, 4.16) | 1.57 (1.01, 2.44) * | .92(.49, 1.70) | 247(0.95, 6.42) | 0.82(0.29, 2.3) | 0.88(0.40, 1.95) |
| Unskilled manual | 0.97 (0.63, 1.51) | 1.77 (1.34, 2.36) * | .95(.38, 2.40) | 1.78(0.48, 6.59) | 0.82 (0.29, 2.32) | 1.25(0.85, 1.83) |
| Others | N/A | N/A | 1.35(.66,2.74) | 2.07(0.74, 5.76) | N/A | N/A |
| Marital status | | | | | | |
| Never in union | 0.17 (0.10, 0.28) * | 0.26 (0.18, 0.38) * | 0.11(0.70, 0.17) * | 0.02(0.01, 0.05) * | 5.35 (4.1, 6.98) * | 0.17(0.09, 0.30) * |
| Currently in union/living with a man | 1.04(0.69, 1.56) | 1.10 (0.79, 1.55) | 1.09(.71, 1.68) | 1.90 (1.12, 3.24) | 15.03 (3.51, 7.2) | 1.01(0.54, 1.88) |
| Formerly in union/living with a man | 1 | 1 | 1 | 1 | 1 | 1 |
| Household wealth quantile | | | | | | |
| Poorest | 0.58(0.38, 0.89) * | 0.82 (0.58, 1.14) | 1.02(.74, 1.40) | 0.62 (0.44, 0.88) * | 1.13 (0.85, 1.49) | 0.84(0.54, 1.33) |
| Poorer | 1.04(0.77, 1.40) | 0.94 (0.68, 1.29) | 0.74(.51, 1.07) | 0.79 (0.56, 1.11) | 0.99 (0.76, 1.28) | 0.89(0.54, 1.39) |
| Middle | 0.69(0.49, 0.98) | 1.10 (0.83, 1.48) | 0. 97(.68, 1.40) | 0.94 (0.67, 1.31) | 0.99 (0.78, 1.24) | 1.05(0.68, 1.61) |
| Richer | 0.69(0.51, 0.92) | 1.09 (0.85, 1.42) | 0. 68(.49, .96) * | 0.85 (0.61, 1.19) | 0.94 (0.76, 1.17) | 0.88(0.59, 1.32) |
| Richest | 1 | 1 | 1 | 1 | 1 | 1 |
| Relationship to the household head | | | | | | |
| Head | 1.65 (0.62, 4.36) | 3.66 (2.32, 5.78) * | 1.10(.53,2.3) | 0.17 (0.05, 0.53) * | 1.20 (0.59, 2.43) | 1.76(0.63, 4.98) |
| Wife | 1.86 (0.70, 4.94) | 3.68 (2.60, 5.12) * | 0. 99(.46, 2.10) | 0.34 (0.105, 1.08) | 0.82 (0.42, 1.60) | 1.94(0.68, 5.52)1 |
| Daughter | 1.43 (0.55, 3.73) | 0.79 (0.54, 1.18) | 0.70(.33, 1.50) | 0.33(0.11, 0.97) * | 0.53 (0.3, 0.95) | 1.05(0.43, 2.56) |
| In-law | 1.93 (0.68, 5.48) | 2.62 (1.71, 4.03) * | 1.13(.45, 2.82) | 0.053(0.006,0.48) * | 0.64 (0.33, 1.23) | 0.289(0.03, 2.50) |
| Other relatives | 1.675 (0.60, 4.67) | 1.51 (0.9, 2.28) | 0.56(.22, 1.40) | 0.54 (0.17,1.71) | 0.73 (0.39,1.32) | 0.62(0.19, 2.05) |
| Not related | 1 | 1 | 1 | 1 | 1 | 1 |

AOR adjusted odds ratio, CI confidence interval

*Shows statistically significant association (p<0.05)

On the other hand, the current study is lower compared to results from a study on the patterns of pregnancy termination among women in Ethiopia, Uganda, and in twenty-seven sub-Saharan countries which were 10.9%, 12%, and 16.5% respectively [40–42]. The difference in the age group of women in the present study, and different study settings might be some of the reasons for the variation of the findings. Furthermore, the variation might be that the current study focused on the reproductive age group of women in the age range of 15–29, whereas

most of the previous studies considered women in the age group of 15–49, which could contribute to the high number of pregnancy terminations.

This study also revealed that Ethiopia had the lowest pregnancy termination rate (4.0%) compared to other countries included in this study. The plausible explanation for this is that Ethiopia is full of preventive culture, norms and religions where pregnancy termination is condemned. In addition, abortion is not legally acceptable in the country except for risky pregnancies or allowed under a range of socioeconomic circumstances [43–45].

In this study, we found that working women had a higher chance of pregnancy termination compared to non-working women which is consistent with previous studies conducted in Ghana [46], Brazil [47], and Nigeria [48]. Working women may have had higher rates of pregnancy termination because of career pursuit.

Pregnancy termination was higher in women aged between 25–29 compared to women aged between 15–19 in all countries. This finding concur with previous studies conducted in Ghana [46], and Nigeria [7], because almost all women of this age group are sexually active [49]. Women with no education were more likely of pregnancy termination compared to women with higher education in Tanzania. This finding was in tandem with previous study conducted in Ethiopia where educated women were less likely to undergo abortions compared to uneducated [31]. This could be due to the fact that uneducated women lack knowledge to manage reproductive health related issues including unplanned pregnancy and abortion.

Household wealth quintile was significantly associated with pregnancy termination among women in Kenya, Burundi and Ethiopia. The study found that women who were in the poorest household wealth quantile in Kenya and Burundi; women who were in the richer household wealth quantile in Ethiopia had lesser odds of pregnancy termination compared to those who were in the richest wealth quintile. Our finding agrees with earlier studies [41, 48, 50, 51]. This is because women from richest wealth quintile have financial capability and can access pregnancy termination services. Ethiopia, just like most of the sub-Sahara African countries, has not fully legalized abortion but allowed under some circumstances [45, 52] thereby making it unaffordable to the richer women compared to the richest ones; it can only be accessed in limited private clinics [53]. Furthermore, the richest women are more civilized, and career driven and may consider pregnancy termination/abortion due to their career pursuit. Additionally, the financial empowerment and better access to safe abortion could contribute to the higher pregnancy termination [54].

In Tanzania and Burundi there was association between relationship to the household head and pregnancy termination. The study showed in Tanzania, the likelihood of having pregnancy termination was higher in household headed by woman, man and in-law compared to household headed by someone not related to the respondents. Women who live with someone who are not related to them might have a less likelihood of getting pregnant since they are hosted and might not have the freedom to do everything they want.

## Conclusion

The result of this study showed that number of women had pregnancy termination. Age group, highest level of education, employment status, marital status, household wealth quantile, and relationship to the household head of the study participants were significantly associated with pregnancy termination. Taking these socio-economic factors into consideration by stakeholders would help tackle the problem. Multisectoral engagement like community leaders, religious leaders should also be there to enhance awareness related to pregnancy termination and related complications. Sexual and reproductive health education targeted to women aged 15–29 also very important to solve the problem.

## Strength and limitation

The use of DHS data set which used a validated questionnaire of DHS MEASURE was the strength of the study while use of a cross sectional study design that cannot describe cause and effect relationship of variables was the limitation of the study. Besides, reporting and recall bias, particularly for age or other retrospective data relying on memory of a past event are some of the limitations of DHs data set.

## Supporting information

**S1 Checklist. STROBE statement checklist of items of a study entitled factors associated with pregnancy termination among women aged 15 to 29 in sub-Saharan Africa.** (DOCX)

**S1 Data. Demographic and Health Survey data sets for six sub-Sahara African countries.** (ZIP)

## Acknowledgments

We thank the Pan African University, Life and Earth Sciences Institute (PAULESI), University of Ibadan, Ibadan, Nigeria) for the opportunity. We would also like to thank the MEASURE DHS Program and ICF International for providing us the permission to use the EDHS data. We also would like to thank staff members of the university for their technical support.

## Author Contributions

**Conceptualization:** Rahel Nega Kassa, Emily Wanja Kaburu, Uduak Andrew-Bassey, Saad Ahmed Abdiwali, Bonfils Nahayo, Ndayishimye Samuel, Joshua Odunayo Akinyemi.

**Data curation:** Rahel Nega Kassa, Emily Wanja Kaburu, Uduak Andrew-Bassey, Saad Ahmed Abdiwali, Bonfils Nahayo, Ndayishimye Samuel, Joshua Odunayo Akinyemi.

**Formal analysis:** Rahel Nega Kassa, Emily Wanja Kaburu, Uduak Andrew-Bassey, Saad Ahmed Abdiwali, Bonfils Nahayo, Ndayishimye Samuel.

**Investigation:** Rahel Nega Kassa, Emily Wanja Kaburu, Uduak Andrew-Bassey, Saad Ahmed Abdiwali, Bonfils Nahayo, Ndayishimye Samuel.

**Methodology:** Rahel Nega Kassa, Emily Wanja Kaburu, Uduak Andrew-Bassey, Saad Ahmed Abdiwali, Bonfils Nahayo, Ndayishimye Samuel, Joshua Odunayo Akinyemi.

**Project administration:** Rahel Nega Kassa, Emily Wanja Kaburu, Uduak Andrew-Bassey, Saad Ahmed Abdiwali, Bonfils Nahayo, Ndayishimye Samuel, Joshua Odunayo Akinyemi.

**Resources:** Rahel Nega Kassa, Emily Wanja Kaburu, Uduak Andrew-Bassey, Saad Ahmed Abdiwali, Bonfils Nahayo, Ndayishimye Samuel, Joshua Odunayo Akinyemi.

**Software:** Rahel Nega Kassa, Emily Wanja Kaburu, Uduak Andrew-Bassey, Saad Ahmed Abdiwali, Bonfils Nahayo, Ndayishimye Samuel, Joshua Odunayo Akinyemi.

**Supervision:** Rahel Nega Kassa, Emily Wanja Kaburu, Uduak Andrew-Bassey, Saad Ahmed Abdiwali, Bonfils Nahayo, Ndayishimye Samuel, Joshua Odunayo Akinyemi.

**Validation:** Rahel Nega Kassa, Emily Wanja Kaburu, Uduak Andrew-Bassey, Saad Ahmed Abdiwali, Bonfils Nahayo, Ndayishimye Samuel, Joshua Odunayo Akinyemi.

**Visualization:** Rahel Nega Kassa, Emily Wanja Kaburu, Uduak Andrew-Bassey, Saad Ahmed Abdiwali, Bonfils Nahayo, Ndayishimye Samuel, Joshua Odunayo Akinyemi.

**Writing – original draft:** Rahel Nega Kassa, Emily Wanja Kaburu, Uduak Andrew-Bassey, Saad Ahmed Abdiwali, Bonfils Nahayo, Ndayishimye Samuel, Joshua Odunayo Akinyemi.

**Writing – review & editing:** Rahel Nega Kassa, Emily Wanja Kaburu, Uduak Andrew-Bassey, Saad Ahmed Abdiwali, Bonfils Nahayo, Ndayishimye Samuel, Joshua Odunayo Akinyemi.

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
