## [Decision Letter · Decision Letter 0]

17 May 2023

PGPH-D-23-00472

Factors Associated with Pregnancy Termination in Sub-Saharan Africa

Dear  Rahel Nega Kassa,

Thank you for submitting your manuscript to PLOS Global Public Health. After careful consideration, we feel that it has merit but does not fully meet PLOS Global Public Health’s publication criteria as it currently stands. Therefore, we invite you to submit a revised version of the manuscript that addresses the points raised during the review process.

We look forward to receiving your revised manuscript.

Kind regards,

Lana Clara Chikhungu, PhD

Academic Editor

Journal Requirements:

1. We noticed you have some minor occurrence of overlapping text with the following previous publication(s), which needs to be addressed:

- https://doi.org/10.17352/ijsrhc.000005

- https://hegelianizm.wordpress.com/2021/03/08/anti-abortion-laws-gender-based-violence-and-marginalization/

In your revision ensure you cite all your sources (including your own works), and quote or rephrase any duplicated text outside the methods section. Further consideration is dependent on these concerns being addressed.

2. We have noticed that you have uploaded Supporting Information files, but you have not included a list of legends. Please add a full list of legends for your Supporting Information files after the references list. 

3. In the online submission form, you indicated that "The Datasets used and/or analyzed during this study are available upon reasonable request from the corresponding author". All PLOS journals now require all data underlying the findings described in their manuscript to be freely available to other researchers, either 1. In a public repository, 2. Within the manuscript itself, or 3. Uploaded as supplementary information.

Additional Editor Comments (if provided):

There is need to pay extra attention to ensure that there are no missing words and the the sentences are coherent. It may be a good idea to hire an editor to edit your manuscript.

Reviewers' comments:

Reviewer's Responses to Questions

**Comments to the Author**

1. Does this manuscript meet PLOS Global Public Health’s publication criteria? Is the manuscript technically sound, and do the data support the conclusions? The manuscript must describe methodologically and ethically rigorous research with conclusions that are appropriately drawn based on the data presented.

Reviewer #1: Partly

Reviewer #2: Partly

2. Has the statistical analysis been performed appropriately and rigorously?

Reviewer #1: Yes

Reviewer #2: No

3. Have the authors made all data underlying the findings in their manuscript fully available (please refer to the Data Availability Statement at the start of the manuscript PDF file)?

Reviewer #1: No

Reviewer #2: Yes

4. Is the manuscript presented in an intelligible fashion and written in standard English?

Reviewer #1: No

Reviewer #2: Yes

5. Review Comments to the Author

Reviewer #1: I suggest that the authors hire a copyeditor.

The authors stated that ʿthe Datasets used and/or analyzed during this study are available upon reasonable request from the corresponding authorʾ. I suggest the author upload the data set as supplementary file.

Abstract (p. 2)

Correct the total number of participants included in this study. The figure quoted is less than the one stated under the methods

Introduction (p. 4,5)

Reference the second sentence of paragraph one.

Write the full meaning of WHO before using the abbreviation for the first time.

Indicate if the figures on the last sentence of the first paragraph are worldwide figures. If yes, how different are these figures from the previous statistics?

I suggest you delete the first sentence of the third paragraph as it has no direct relation to the topic.

Mention the nations with 25% of unsafe abortion.

Include two or more references since you mentioned studies in sentence three of the fourth paragraph.

Is the 48 abortions per 1000 women of reproductive age an annual figure? Which country’s statistic is this?

Rephrase the third sentence of paragraph 5. The sentence means that abortion causes 454 deaths per 100,000 live births in Tanzania. p.5

Methods

Population (p.6)

Write the full meaning of IR before using the abbreviation.

Justify why you chose women aged 15 to 29.

Results

Sociodemographic characteristics (p. 7-9)

Correct the narrative on the age of the respondents in Kenya. In Kenya majority of participants were women aged 15 -19(35%) and 25 -29 (35%) respectively.

Table 2: Recalculate the total sample for employment status for Ethiopia. The sample is more than n=9246

Check the total sample for employment status, marital status and Household wealth quantile for Kenya. The figures are not up to n=8257

Check the total sample for marital status for Nigeria. It is more than n=22470

Prevalence (p. 10)

The total number of women aged 15 -29 who were included in this study are less than the figure stated.

Include the overall prevalence of pregnancy termination in the study population.

Delete the last sentence under the prevalence since the figures are clearly stated in table 2.

Factors associated with pregnancy termination (p.10,11)

Include the results of the bivariate analysis in the narrative.

Correct the arrangement of the results on age and pregnancy termination for Burundi and Tanzania in the narrative; Burundi (AOR: 2.4 95%CI 1.5, 3.8), Tanzania (AOR: 2.7 95%CI 2.02, 3.57) on page 11 as women who were in the age group of 25-29 were more likely to have a higher chance of pregnancy termination compared to women aged 15-19 in Tanzania than Burundi.

Discussion (p. 15-17)

Clarify the fourth and fifth sentences of the first paragraph of the discussion; ʿThis finding is lower compared to the study done among young women in another study that was done on the patterns of pregnancy termination, Uganda, and in twenty-seven sub-Saharan countries which were 10.9%, 12%, and 16.5% respectively(36-38). The reason for this might be that the number of study participants in the former studies used a larger sample size which was DHS data from 50 countriesʾ

You mentioned in Uganda and 27 sub-Saharan countries however, you stated three different prevalence instead of two. Also, revisit the explanation for the higher prevalence recorded on the fifth sentence since there was no prior mention of DHS data from 50 countries in the manuscript.

Discuss further the reasons for the higher rate of abortion in sub-Saharan Africa as compared to well developed countries.

Also, discuss studies that reported lower abortion rate among your study participants and explain the findings.

Discuss thoroughly why women with no education are more likely to terminate their pregnancies. P. 16

Reference the last three sentences of the sixth paragraph and the last paragraph on page 16 and 17 of the discussion.

Conclusion (P. 17)

Include the main contribution of the study to your field of practice or academic knowledge.

Strength and limitation (p. 17)

Include limitations of DHS data set.

References (p. 20-22)

Include the Doi link

PDF attachment (ethics)

One of the PDF ethics attachment titled ʿMaternal Health Intervention in Reducing Maternal Mortality in Tanzania": is different from your research topicʿ clarify.

Reviewer #2: Comment 1: Statistically this can't be claimed: Hence, withthe rest of the included countries, the findings of this study can be generalized to the rest of the Sub-Saharan African countries.

Comment 2: Mention the version of DHS questionnaire for each survey and discuss the differences, if any. "The DHS uses three core questionnaires adapted from the MEASURE DHS project."

Comment 3: How the sampling design was controlled during analysis? Mention in clearly with the we use the v005 sampling weight variable for bivariate as well as multi-variate analysis. "DHS collects data from household

samples using a two-phase stratified cluster design"

Comment 4: Dependent variable - is this variable includes the forced abortion as well? Please review.

Comment 5: Bi-variate analysis p-value must be set at 0.25 to include not only the statistical significant variables but to include the contextual important variable too. Read - Model Building chapter from Applied Logistic Regression book.

Comment 6: The statistically significant independent variable in the bi-variate model must be checked for multicollinearity with variance inflation factor before taking them into the final model. The appropriate level of VIF for logistic regression must be used to rule out the confounding effect.

Comment 7:

6. PLOS authors have the option to publish the peer review history of their article (what does this mean?). If published, this will include your full peer review and any attached files.

**Do you want your identity to be public for this peer review?** For information about this choice, including consent withdrawal, please see our Privacy Policy.

Reviewer #1: No

Reviewer #2: **Yes: **Shital Bhandary

---

## [Decision Letter · Decision Letter 1]

14 Sep 2023

PGPH-D-23-00472R1

Factors Associated with Pregnancy Termination in Sub-Saharan Africa

Dear Dr. Kassa,

Thank you for submitting your manuscript to PLOS Global Public Health. After careful consideration, we feel that it has merit but does not fully meet PLOS Global Public Health’s publication criteria as it currently stands. Therefore, we invite you to submit a revised version of the manuscript that addresses the points raised during the review process.

We look forward to receiving your revised manuscript.

Kind regards,

Lana Clara Chikhungu, PhD

Academic Editor

Journal Requirements:

2. We have noticed that you have a list of Supporting Information legends in your manuscript. However, there are no corresponding files uploaded to the submission. Please upload them as separate files with the item type 'Supporting Information'. 

Additional Editor Comments (if provided):

Reviewers' comments:

Reviewer's Responses to Questions

**Comments to the Author**

1. If the authors have adequately addressed your comments raised in a previous round of review and you feel that this manuscript is now acceptable for publication, you may indicate that here to bypass the “Comments to the Author” section, enter your conflict of interest statement in the “Confidential to Editor” section, and submit your "Accept" recommendation.

Reviewer #1: (No Response)

Reviewer #2: All comments have been addressed

2. Does this manuscript meet PLOS Global Public Health’s publication criteria? Is the manuscript technically sound, and do the data support the conclusions? The manuscript must describe methodologically and ethically rigorous research with conclusions that are appropriately drawn based on the data presented.

Reviewer #1: Yes

Reviewer #2: Yes

3. Has the statistical analysis been performed appropriately and rigorously?

Reviewer #1: Yes

Reviewer #2: No

4. Have the authors made all data underlying the findings in their manuscript fully available (please refer to the Data Availability Statement at the start of the manuscript PDF file)?

Reviewer #1: Yes

Reviewer #2: Yes

5. Is the manuscript presented in an intelligible fashion and written in standard English?

Reviewer #1: Yes

Reviewer #2: Yes

6. Review Comments to the Author

Reviewer #1: I suggest the authors hire a copy editor.

Introduction

Put a full stop at the end of the sentence; In Sub-Saharan Africa ………..preventable(6). Third sentence of the second paragraph. p. 3

Put a full stop at the end of the sentence; Thirteen percent …………… unsafe abortion(13).

second sentence of the third paragraph. p. 3

Use a smaller letter to start the word ‟the” aim……. Last sentence of the last paragraph. p. 4

Methods

These countries were selected because of the high prevalence of abortions in the regions where they are found in. I suggest you write it as; These countries were selected because of the high prevalence of abortions in the region” p. 4

Socio-demographic Characteristics

Correct the sentence; In Kenya, Tanzania, Ethiopia, Burundi, Nigeria and Rwanda majority of

the respondents were aged between 15-19. In Kenya majority 6100 (34.5)of the participants were between the ages of 25-29 not 15-19. The first sentence of the socio-demographic factors. p. 6

Factors associated with pregnancy termination

Correct the sentence on the third paragraph: women living in rural areas compared to women living in rural. p. 10

Correct the sentence: women with a primary and secondary level are ............…….. in Ethiopia. p. 14. The reported findings are for no education and primary education respectively, not primary and secondary education.

Under marital status, correct the figure 72% (AOR: 0.28 95% 0.18, 0.41) for women in Nigeria who had never been in a union. The figure reported under table 3 was 5.35 (4.1, 06.98). p. 15,17

Correct the results: Burundi, women who had relationship to the household ………............. in law 99% (AOR: 0.052, 95% CI 0.006, 0.48) less likely to terminate pregnancy. The 99% should be 95%. p. 15

Also correct the word similarly on the last sentence of the result before table 3. p. 15

Discussion

Remove the almost in front of 6.25% on the second sentence of the first paragraph second. p. 19. The prevalence was 6.25%.

Further explain the statement; European countries are very developed with educated and civilized women who could forecast the consequences of pregnancy termination. Include reference as well. p. 19

Include references for these sentences; Besides, the low socioeconomic status among women in the sub-Saharan countries might lead …………………… compared to the well-developed ones. p. 19

Reference the sentences; The plausible explanation for this is that Ethiopia ………........... except for risky pregnancies. p. 19

Reference the sentence; Findings of this study ……………….................... because almost all women of this age group are sexually active. p. 20.

Reference these sentence; This may be related to the fact ………………….......... high salary/payment due to their low level of education. p. 20

Clarify the sentence; In almost 50% of countries included in this study, women with only primary education were more likely to undergo an abortion. p. 20. Are you referring to your current study or another study? Because in the current study you included only 6 countries.

Clarify the last part of the sentence on paragraph 4; maybe the programs may not include related courses for those who attended only primary education. What related courses are you referring to? p. 20

Ethiopia just like most of the Sub-Saharan African countries has not legalized abortion(50). This is not true. In Ethiopia abortion is allowed by law based on certain conditions. p. 21

The last paragraph of the discussion; join likelihood, it is one word. p. 21

Reviewer #2: Previous Comment 6: The statistically significant independent variable in the bi-variate model

must be checked for multicollinearity with variance inflation factor before taking them

into the final model. The appropriate level of VIF for logistic regression must be used to

rule out the confounding effect.

Answer to this comment: Multicollinearity between covariates was checked using the variance inflation factor (VIF) and VIF values greater than 10 indicate the existence of multicollinearity (This is ok for linear regression model only as it used Ordinary Least Square method so this is not correct for the logistic regression as it uses Maximum Likelihood method)

You must use VIF cut-off of 2 for logistic regression as per the book I had recommended you earlier. This is very important as we don't want the confounding variable/s in the final multivariate model. So, kindly re-run the analysis and submit again.

7. PLOS authors have the option to publish the peer review history of their article (what does this mean?). If published, this will include your full peer review and any attached files.

**Do you want your identity to be public for this peer review?** For information about this choice, including consent withdrawal, please see our Privacy Policy.

Reviewer #1: No

Reviewer #2: **Yes: **Shital Bhandary

---

## [Editor Report · Decision Letter 2]

16 Oct 2023

PGPH-D-23-00472R2

Factors Associated with Pregnancy Termination in Sub-Saharan Africa

Dear Dr. Nega Kassa

Thank you for submitting your manuscript to PLOS Global Public Health. After careful consideration, we feel that it has merit but does not fully meet PLOS Global Public Health’s publication criteria as it currently stands. Therefore, we invite you to submit a revised version of the manuscript that addresses the key issue of poor english, incomprehensible conclusions and inconsistency in the use of the tern Sub Saharan Africa regions.  The article needs to be edited so that it is presented in an intelligible fashion.  You are requested to seek independent editorial help before submitting a revision. These services can be found on the web using search terms like “scientific editing service” or “manuscript editing service.”

We look forward to receiving your revised manuscript.

Kind regards,

Lana Clara Chikhungu, PhD

Academic Editor

Journal Requirements:

Additional Editor Comments (if provided):

Despite the improvements that the authors have made, the quality of the paper still needs significant improvements. The big issue is the standard of communication. The whole paper needs re writing. I suggest that you send the paper to an editor with good English writing skills but also find someone to check the presentation of ideas.
---

## [Decision Letter · Decision Letter 3]

19 Feb 2024

PGPH-D-23-00472R3

Factors Associated with Pregnancy Termination in Selected sub-Sahara African Countries

Dear Dr. Kassa,

Thank you for submitting your manuscript to PLOS Global Public Health. After careful consideration, we feel that it has merit but does not fully meet PLOS Global Public Health’s publication criteria as it currently stands. Therefore, we invite you to submit a revised version of the manuscript that addresses the points raised during the review process.

The manuscript has been evaluated by two reviewers, and their comments are available below.

The reviewers are both positive towards your revised manuscript and have only recommended minor revisions. Could you please carefully revise the manuscript to address all comments raised?==============================

We look forward to receiving your revised manuscript.

Kind regards,

Johanna Pruller, Ph.D.

PLOS Staff Editor

Journal Requirements:

2. We have noticed that you have cited Table 4 in the manuscript file but there are no corresponding tables in the manuscript. Please amend your manuscript to include this table, noting that tables should not be uploaded as individual files.

Additional Editor Comments (if provided):

Reviewers' comments:

Reviewer's Responses to Questions

**Comments to the Author**

1. If the authors have adequately addressed your comments raised in a previous round of review and you feel that this manuscript is now acceptable for publication, you may indicate that here to bypass the “Comments to the Author” section, enter your conflict of interest statement in the “Confidential to Editor” section, and submit your "Accept" recommendation.

Reviewer #1: All comments have been addressed

Reviewer #2: All comments have been addressed

2. Does this manuscript meet PLOS Global Public Health’s publication criteria? Is the manuscript technically sound, and do the data support the conclusions? The manuscript must describe methodologically and ethically rigorous research with conclusions that are appropriately drawn based on the data presented.

Reviewer #1: Yes

Reviewer #2: Partly

3. Has the statistical analysis been performed appropriately and rigorously?

Reviewer #1: Yes

Reviewer #2: No

4. Have the authors made all data underlying the findings in their manuscript fully available (please refer to the Data Availability Statement at the start of the manuscript PDF file)?

Reviewer #1: Yes

Reviewer #2: Yes

5. Is the manuscript presented in an intelligible fashion and written in standard English?

Reviewer #1: Yes

Reviewer #2: Yes

6. Review Comments to the Author

Reviewer #1: The quality of the manuscript has improved substantially.

Results

Sociodemographic

In Tanzania majority (36.9%) of the participants were rather working in the agriculture sector. p 6

Discussion

Rephrase the last sentence of paragraph one. p14

Delete 'which were' after Norway in the first sentence of paragraph two. P 14

Reference the last but one sentence of paragraph five ‘ it can only be accessed in limited private clinics’ p 15

Also reference the last sentence of paragraph five ‘The richest women are more civilized, and career driven and may consider pregnancy termination/abortion due to their career pursuit’. p 15

Reviewer #2: The authors have mentioned that they have used p-value of 0.25 in the bi-variate analysis as cut-off to include independent variables in the final model but this is not seen in any Tables presented not it is discussed in the result section. Further, they also mentioned in the method section that VIF cut-off of 2 was used to take variables with p-value less than 0.25 in the final model but the results/tables do not show it and even none of the text in the result section described it. Thus, it is recommended to add a table with bi-variate analysis results separately, like Table 3, in the manuscript and show the actual p-values there for all the countries. Table 3 or separate section must show/mention the VIF values for each country for each variables included in the final model so that the Table 3 can be taken as the final result and conclusions can also be taken as valid and reliable.

It is not clear what this statement means in the result section: "After running the significantly associated variables in multivariate logistic regression except for type of residence and sex of household, the rest variables were significantly associated with the outcome variable at least in one of the six countries."

7. PLOS authors have the option to publish the peer review history of their article (what does this mean?). If published, this will include your full peer review and any attached files.

**Do you want your identity to be public for this peer review?** For information about this choice, including consent withdrawal, please see our Privacy Policy.

Reviewer #1: No

Reviewer #2: **Yes: **Shital Bhandary

---

## [Decision Letter · Decision Letter 4]

11 Apr 2024

Factors Associated with Pregnancy Termination in six sub-Saharan African Countries

PGPH-D-23-00472R4

Dear Dr. Kassa,

We are pleased to inform you that your manuscript 'Factors Associated with Pregnancy Termination in six sub-Saharan African Countries' has been provisionally accepted for publication in PLOS Global Public Health.

Best regards,

Dandara de Oliveira Ramos, PhD

Academic Editor

Reviewer Comments (if any, and for reference):

Reviewer's Responses to Questions

**Comments to the Author**

1. If the authors have adequately addressed your comments raised in a previous round of review and you feel that this manuscript is now acceptable for publication, you may indicate that here to bypass the “Comments to the Author” section, enter your conflict of interest statement in the “Confidential to Editor” section, and submit your "Accept" recommendation.

Reviewer #1: All comments have been addressed

Reviewer #2: All comments have been addressed

2. Does this manuscript meet PLOS Global Public Health’s publication criteria? Is the manuscript technically sound, and do the data support the conclusions? The manuscript must describe methodologically and ethically rigorous research with conclusions that are appropriately drawn based on the data presented.

Reviewer #1: Yes

Reviewer #2: Yes

3. Has the statistical analysis been performed appropriately and rigorously?

Reviewer #1: Yes

Reviewer #2: Yes

4. Have the authors made all data underlying the findings in their manuscript fully available (please refer to the Data Availability Statement at the start of the manuscript PDF file)?

Reviewer #1: Yes

Reviewer #2: Yes

5. Is the manuscript presented in an intelligible fashion and written in standard English?

Reviewer #1: Yes

Reviewer #2: Yes

6. Review Comments to the Author

Reviewer #1: (No Response)

Reviewer #2: All the technical comments including the multicollinearity has been addressed now. However, the table numbers with VIF values must be re-assessed.

7. PLOS authors have the option to publish the peer review history of their article (what does this mean?). If published, this will include your full peer review and any attached files.

**Do you want your identity to be public for this peer review?** For information about this choice, including consent withdrawal, please see our Privacy Policy.

Reviewer #1: No

Reviewer #2: **Yes: **Shital Bhandary
